

# Modelling long-term, large-scale sediment storage using a simple sediment budget approach

Victoria Naipal[1], Christian Reick[1], Kristof Van Oost[2], Thomas Hoffmann[3], and Julia Pongratz[1]

[1]Department of Land in the Earth System, Max Planck Institute for Meteorology, Hamburg, Germany
[2]Université catholique de Louvain, TECLIM – Georges Lemaître Centre for Earth and Climate Researh, Louvain-la-Neuve, Belgium
[3]Department of Geography, University of Bonn, Bonn, Germany

*Correspondence to:* V.Naipal (victoria.naipal@mpimet.mpg.de)

**Abstract.**

Currently, the anthropogenic disturbances to the biogeochemical cycles remain unquantified due to the poor representation of lateral fluxes of carbon and nutrients in Earth System Models (ESMs) that couple the terrestrial and ocean systems. Soil redistribution plays an important role in the transport of carbon and nutrients between terrestrial ecosystems, however, quantification of soil redistribution and its effects on the global biogeochemical cycles is missing. This study aims at developing new tools and methods to represent soil redistribution on a global scale, and contribute to the quantification of anthropogenic disturbances to the biogeochemical cycles. We present a new large-scale coarse resolution sediment budget model that is compatible with ESMs. This model can simulate spatial patterns and long-term trends in soil redistribution in floodplains and on hillslope, resulting from external forces such as climate and land use change. We applied this model on the Rhine catchment using climate and land cover data from the Max Planck Institute Earth System Model (MPI-ESM) for the last millennium (850 - 2005AD). Validation is done using observed Holocene sediment storage data and observed scaling relations between sediment storage and catchment area from the Rhine catchment. We found that the model reproduces the spatial distribution of floodplain sediment storage and the scaling relationships for floodplains and hillslopes as found in observations. The exponents of the scaling relationships can be modified by changing the spatial distribution of erosion or by changing the residence time for floodplains. However, the main feature of the scaling behavior, which is that sediment storage in floodplains increases stronger with catchment area than sediment stored on hillslopes, is not changed. Based on this we argue that the scaling behavior is an emergent feature of the model and mainly dependent on the underlying topography. Additionally, we identified that land use change explains most of the temporal variability in sediment storage for the last millennium in the Rhine catchment.



## 1 Introduction

Soil erosion by rainfall and the resulting sediment deposition and transport (soil redistribution) play an important role in the mineralization and sequestration of soil carbon and the loss of carbon and nutrients from ecosystems (Van Oost et al., 2007). On the one hand, mineralization of soil carbon at eroding sites and during transport can lead to fluxes of greenhouse gasses (Lal, 2003; Van Oost et al., 2007; Lal, 2005). On the other hand, the transport of carbon and nutrients from a terrestrial

ecosystem can result in either sequestration of carbon at deposition sites (Stallard; Van Oost et al., 2007), or significant lateral fluxes of carbon and nutrients (Van Oost et al., 2007; Quinton et al., 2010).

Recent evidence demonstrated that human activities, such as land use change, have accelerated soil erosion and altered the lateral transport of carbon and nutrients (Regnier et al., 2013; Stallard; Bauer

et al., 2013; Le Quéré et al., 2013). For example, Regnier et al. (2013) estimated that human activities increased the carbon flux to inland waters by as much as $1.0\,\mathrm{Pg\,carbon\,year^{-1}}$. However, the effect of soil redistribution on the vertical and lateral fluxes of carbon and nutrients on a global scale is still unknown. This complicates the quantification of anthropogenic disturbances of the biogeochemical cycles.

Data on large to global scale soil redistribution rates are scarce to non-existing. There exist several modelling approaches to estimate global soil erosion rates (Yang et al., 2003; Ito, 2007; Montgomery, 2007; Doetterl et al., 2012; Naipal et al., 2015). These modelling approaches mainly address the soil detachment process only, and do not simulate the dynamics of sediment by ignoring processes such as sediment deposition and transport. There is, to our knowledge, no globally applicable model that

can explicitly simulate soil redistribution, which is a result of the sediment dynamics in a landscape, for the past, present and future. The lack of such kind of large-scale models on soil redistribution substantially limits the understanding of the relative importance of the various effects of soil erosion and related processes on the global biogeochemical cycles.

The holistic understanding of the interaction and linkages between soil erosion, deposition and trans-

port, can be addressed using the sediment budget approach (Walling and a.L. Collins, 2008). Slaymaker (2003) defined the sediment budget as a mass-balance-based approach where the mass of sediments, water or nutrients is conserved in the considered system so that the net increase in storage is equal to the excess of inflow over outflow of the conserved quantity. However, long-term large-scale sediment budgets are very scarce to non existing. Sediment budgets that have been constructed previ-

ously range from small catchments (Verstraeten and Poesen, 2000; Walling et al., 2001) to large river catchments (Milliman and Meade, 1983; Ludwig and Probst, 1998; Syvitski et al., 2003; Slaymaker, 2003). However, these sediment budgets are usually for present day only as they are mostly based on measurements using methods such as sediment tracing or fingerprinting. Also, most of these studies only focus on the sediment delivery from a catchment. These studies are therefore of limited use for

assessing the spatial distribution of sediment sources and storage or in predicting long-term sedi-



ment yields. Considering explicitly the spatial distribution of these variables within a catchment is not only essential for a proper land management strategy to combat land degradation, but also for a detailed assessment of how erosion and sediment transport interact with the carbon and nutrient cycles. For example, it is important to distinguish between sediment related processes in floodplain

and hillslope systems (Hoffmann et al., 2013). Human activities usually lead to a stronger increase in sediment deposits on hillslopes compared to floodplains, and an overall decreased export of sediment out of a catchment, despite increased soil erosion (de Moor and Verstraeten, 2008). In this way, sediments stored in floodplains and hillslopes over long timescales can significantly delay or alter the human induced changes to the carbon and nutrient cycles (Hoffmann et al., 2013). This indicates

that there is a need for long-term sediment budgets, as they can provide essential information on the forces behind sediment, carbon and nutrient fluxes in a catchment such as human activities and climate change.

There is thus a need for spatially explicit models that can simulate long-term sediment budgets. There exist different spatial models of suspended sediment flux that also consider the soil redistri-

bution or sediment dynamics in a catchment (Merritt et al., 2003; de Vente and Poesen, 2005; Ward et al., 2009). However, many of them are developed to simulate single events or require input data that is not available for large spatial scales (Wilkinson et al., 2009). There are also partly empirical models which can operate on catchment scale such as the WATEM/SEDEM model, which is used to predict hillslope sediment storage and sediment yields (de Moor and Verstraeten, 2008; Nadeu et al.,

2015). Or such as the suspended sediment model from Wilkinson et al. (2009) that also simulates some other processes such as floodplain deposition, gully and riverbank erosion. However, these models are not compatible for a global scale application as they require parameters for which data is not available on a global scale and these type of models also need to be calibrated to measured sediment yields of the studied area (Van Rompaey et al., 2001). Pelletier (2012) proposed a global

applicable model for long-term suspended sediment discharge, where he used various environmental controlling parameters to simulate soil detachment and sediment transport. However, in his study he mainly focuses on the sediment discharge and delivery of catchments and his model does not take into account the full dynamics of sediment in a catchment, which would also include the spatial distribution of sediment deposition and storage in the different reservoirs of a catchment. Furthermore,

he does not consider land use change and thus his approach is limited to natural catchments only.

The land components of Earth System Models (ESMs) are the main tools to investigate the terrestrial carbon cycle and land use and land cover change (LULCC). They mainly represent the effects of fossil fuels and land use change on the carbon cycle and the resulting carbon flux between soil and the atmosphere (Regnier et al., 2013). However, they ignore the lateral carbon fluxes associated

with soil redistribution and, therefore, miss an important aspect of the coupling between land and the ocean (Regnier et al., 2013; Oost et al., 2012). Including soil redistribution processes in ESMs would facilitate this coupling and create the possibility to study the full interactions and feedbacks




between the soil and the biogeochemical cycles on a global scale.

The overall aim of this study is to contribute to the quantification of the anthropogenic impact on

lateral fluxes of carbon and nutrients through the representation of sediment dynamics and associated lateral fluxes in global ESMs. Therefore, we present a new large-scale sediment budget model that is able to simulate spatial patterns and long-term trends of soil erosion, sediment deposition and sediment storage, based on climate and land use changes. Compatibility of this new model with ESMs is important for a future extension of the model to include the carbon and nutrient cycling.

The specific objective of this study is to present and evaluate the new model for the non-Alpine part of the Rhine catchment using the environment of ESMs. The choice of the Rhine catchment is based on the fact that it is the only large catchment with a long land use history for which we had long-term sediment storage data available. For the validation of the model we used scaling relationships between sediment storage and catchment area found from observations for the non-Alpine part of the

Rhine catchment by Hoffmann et al. (2013). The scaling relationships are an important criteria for the sediment budget model, as they represent the overall main behavior of sediment in a catchment as function of catchment area. These relationships can thus function as a simple test for the spatial variability of stored sediment that is modelled with a large-scale coarse resolution sediment budget model. We use the model to quantify the spatial variability of floodplain and hillslope sediment stor-

age for the Rhine catchment, and its dependence on climate change and land use change during the last millennium (850-2005AD). Finally, we discuss the main challenges in modelling large-scale, long-term soil erosion and soil redistribution and future perspectives for application in ESMs.

## 2 Methods

### 2.1 Basic model concept

The main purpose of the sediment budget model presented here, is to estimate large-scale long-term floodplain and hillslope sediment storage and lateral fluxes of sediment. The model should therefore be spatially explicit and capable of estimating erosion, deposition and sediment transport processes. Furthermore, we want to differentiate between floodplain and hillslope sediment storage for a better quantification of the impact of human activities in a catchment. Therefore, we use a

grid cell based approach where we assume that each grid cell contains a floodplain and hillslope reservoir. Before we can define a model that satisfies the above mentioned conditions we have to make some basic assumptions. Firstly, as it is difficult to disentangle the floodplains and hillslopes in available soil data sets, we assume that each grid cell contains both a hillslope and a floodplain reservoir. When estimating large-scale sediment storage with the aim of predicting fluxes of carbon

in the future, the focus is to get the large-scale spatial patterns right, rather than accurate numbers for the sediment storage. Secondly, we assume that the deposition and sediment transport behave differently between the floodplain and hillslope reservoirs on the timescale of the last millennium.



Thirdly, erosion is considered to mainly take place on hillslopes, where part of the eroded sediment is directly transported from hillslopes and deposited in the floodplains.

The underlying model framework (Fig. 1a) that consists out of the erosion, deposition and sediment transport modules, is based on the sediment mass-balance method. The change in sediment storage ($M$) within a certain unit of time and space is given by the difference between sediment input and sediment output (Slaymaker, 2003). For sediment stored in floodplains ($M_a$), this leads to:

$$\frac{dM_a}{dt} = D_a - L \qquad (1)$$

Here, $D_a$ is the sediment deposition rate in floodplains, and L is the sediment loss. Equation 1 can be approximated by the following as function of time:

$$\frac{dM_a}{dt} = D_a(t) - k * Ma(t) \qquad (2)$$

where $D_a(t)$ is the time-dependent input rate in the model, which is independent from $M_a(t)$. $k * M_a(t)$ is the loss term in the floodplain reservoir, and $k$ is the specific rate for floodplains.

The specific rate is the inverse of the residence time ($1/\tau$) for floodplain sediment, which is defined as the time (in years) a soil particle stays in the floodplain reservoir of a certain grid cell. $\tau$ is assumed to be independent of time for timescales in the order of several thousands of years, and is assumed to increase exponentially with catchment area, where the catchment area is represented by the weighted flow-accumulation:

$$\tau = e^{\frac{(FlowAcc - a_\tau)}{b_\tau}} \qquad (3)$$

$a_\tau$ and $b_\tau$ are residence time constants and $FlowAcc$ is the flow-accumulation. Flow-accumulation is defined as the number of grid cells upstream that flow into a certain grid cell. As each grid cell represents a certain catchment area, the value of $\tau$ will be dependent on the location of the grid cell in the catchment. The presented relationship between $\tau$ and catchment area in equation 3 is based

on the fact that large catchment areas are usually characterized by low slopes which mainly result in a low connectivity that makes the system capable of storing sediment for a long time. The opposite is true for small catchment areas, where the connectivity is usually high and the sediment in these systems will therefore have short residence times (Hoffmann, 2015).

The deposition rate ($D_a$) in the floodplain reservoir can be defined as a certain fraction of the erosion

rate. In this way equation 2 can be rewritten as:

$$\frac{dM_a}{dt} = f * E - \frac{M_a(t)}{\tau} \qquad (4)$$





Where $f$ is the dimensionless floodplain deposition fraction ranging between 0 and 1, and $E$ is the erosion rate in $\mathrm{t\,ha^{-1}\,year^{-1}}$.

The erosion rate is computed according to the adjusted Revised Universal Soil Loss Equation (RUSLE) model (Naipal et al., 2015), which computes annual averaged rill and interril erosion rates and is formulated as a product of a rainfall erosivity factor ($R$, $\mathrm{MJ\,mm\,ha^{-1}\,h^{-1}\,yr^{-1}}$), a slope steepness factor ($S$,dimensionless), a soil erodibility factor ($K$, $\mathrm{t\,ha\,h\,ha^{-1}\,MJ^{-1}\,mm^{-1}}$), and a land cover factor ($C$,dimensionless):

$$E = S * R * C * K \tag{5}$$

The underlying RUSLE model stems from the original Universal Soil Loss Equation (USLE) model developed by USDA (USA Department of Agriculture), which is based on a large set of experiments on soil loss due to water erosion from agricultural plots in the United States (Renard et al., 1997). These experiments covered a large variety of agricultural practices, soil types and climatic conditions, making it a potentially suitable tool on a regional to global scale.

In the adjusted RUSLE model, as presented above, the effects of the slope-length ($L$ factor) and support practice ($P$ factor) are excluded. In the original RUSLE model (Renard et al., 1997), these factors are part of the model, however, on large to global scale there is too little data available on these factors. Including them in the model would only result in additional uncertainties, while we try to keep the model simple, to be able to capture and quantify the main processes and drivers behind large-scale sediment mobilization. We do, however, agree that leaving these two factors out could introduce some biases in erosion rates, especially in agricultural areas.

The floodplain deposition fraction is calculated by a simple growth function where deposition is a function of the mean topographical slope and the main land cover type in a grid cell:

$$f = a_f * e^{\{b_f * \frac{\theta}{\theta_{max}}\}} \tag{6}$$

where $a_f$ and $b_f$ are constants for deposition and dependent on the land cover type, and $\theta$ is the average percent slope on a 5 arcmin resolution grid. $\theta_{max}$ is the maximum percent slope. An increase in the overall average slope of a grid cell leads to a larger transport of eroded soil from the hillslopes to the floodplains. This results in an increase in deposition rate to the floodplain reservoir of that specific grid cell. Hereby we consider in equation 6 that this increase is exponential. For a natural landscape we assume a good 'sediment connectivity' between hillslopes and the floodplain in a grid cell. In natural landscapes the sediment connectivity is largely based on the vegetation density and morphological structures (Gumiere et al., 2011; Bracken and Croke, 2007). To keep the model simple we do not adapt these parameters to the complexity of natural landscapes, but rather differentiate between the deposition rates in natural and agricultural landscapes, assuming that the



sediment connectivity differs fundamentally between these landscapes. Several recent studies (Hoff-
mann et al., 2013; de Moor and Verstraeten, 2008; Gumiere et al., 2011) showed that a large part of
the eroded sediment is deposited and stored directly on the hillslopes where agricultural activities
take place. Agricultural activities that use anthropogenic structures, reduce the sediment transport
from hillslopes to the floodplains (Gumiere et al., 2011). In this way, the stored hillslope sediment

is disconnected from the fluvial system on timescales of 100 to a few 1000 years. Based on this, we
assume that for agricultural land (crop and pasture) and grassland the sediment connectivity is dis-
turbed. A bad sediment connectivity will result in a larger fraction of eroded soil that remains on the
hillslopes compared to the fraction that flows along the hillslopes and is deposited in the floodplains.
For natural landscapes we assume a better sediment connectivity, meaning that a larger fraction of

the eroded soil will be deposited in the floodplains compared to the fraction that remains on the
hillslope. Here we ignore morphological conditions that can cause deconnectivity in the landscape.
After calculating erosion and deposition, the sediment is transported between the grid cells based on
a multiple flow sediment routing scheme such as presented by Quinn et al. (1991) (Fig. 1b). In the
multiple flow routing scheme the weight ($W$, dimensionless), which specifies the part of the flow

that comes in from a neighboring grid cell, is calculated as:

$$W_{(i+k,j+l)} = \frac{\theta_{(i+k,j+l)} * c_{(i+k,j+l)}}{\sum_{k,l=-1}^{1} [\theta_{(i+k,j+l)} * c_{(i+k,j+l)}]} \tag{7}$$

where $c$ is the contour length and is respectively, 0.5 in the cardinal direction and 0.354 in the
diagonal direction. $(i,j)$ is the grid cell in consideration where $i$ counts grid cells in the latitude
direction and $j$ in the longitude direction. $i+k$ and $j+l$ specify the neighboring grid cells where $k$

and $l$ can be either -1, 0 or 1. $\theta$ is calculated here as:

$$\theta_{(i+k,j+l)} = \frac{h_{(i,j)} - h_{(i+k,j+l)}}{d} \tag{8}$$

where, $h$ is the elevation in meters derived from a digital elevation model, $d$ is the grid size in
meters.

The floodplain sediment storage rate ($t\,ha^{-1}\,year^{-1}$) of a grid cell $(i,j)$ is then a function of the

deposition rate in that grid cell, the loss from that grid cell and the incoming sediment from the
neighboring grid cells, and is calculated at each time step $t$ as:

$$M_{a(i,j)_{t+1}} = M_{a(i,j)_t} + [f_t * E_t - \frac{M_{a(i,j)_t}}{\tau_{(i,j)}}] + \sum_{k,l=-1}^{1} [\frac{M_{a(i+k,j+l)_t}}{\tau_{(i+k,j+l)}} * W_{(i+k,j+l)}] \tag{9}$$

For hillslopes the change in sediment storage is assumed to be equal to the input rate, because
we assume that the stored hillslope sediment has an infinite residence time on the timescale of the

last millennium (Eq.10). This means that the hillslope sediment storage will increase linearly with



time (Eq.11). The hillslope sediment deposition rate ($D_c$) is here defined as the remaining part of the eroded soil that has not be been transferred to the floodplain directly ($1$-$f$). The equations for the hillslope sediment storage rate ($M_c$, $\mathrm{t\,ha^{-1}\,year^{-1}}$) are represented by:

$$\frac{dM_c}{dt} = D_c = (1 - f) * E \tag{10}$$

$$M_{c(i,j)_{t+1}} = M_{c(i,j)_t} + (1 - f_t) * E_t \tag{11}$$

### 2.2 Model implementation and parameter estimation

The resolution of the sediment budget model is 5 arcmin. We chose this particular model resolution, because we assume that this is the optimal resolution when considering that each grid cell contains a floodplain and hillslope fraction. Here, a higher resolution could lead to cases where this assumption

is not met. Also, the 5 arcmin resolution fits well with the resolution of the adjusted RUSLE model. The sediment budget model uses climate and land cover data from simulations of MPI-ESM that have been performed under the Coupled Model Intercomparison Project Phase 5 (CMIP5) framework (Hurrell and Visbeck, 2011; Taylor et al., 2009). As this data was given at a resolution of approximately 1.875 degrees, we had to downscale the data to the resolution of the sediment bud-

get model. For the period 1850-2005AD three ensemble members from MPI-ESM (r1i1p1, r2i1p1, r3i1p1) were available, while for the period 850-1850AD only one ensemble member (r1i1p1) was available. This data existed on a 6 hourly, monthly or yearly time step for the last millennium. Calculation of soil erosion according to the adjusted RUSLE model is mostly based on the methods presented in the study of Naipal et al. (2015). However, the calculation of the $R$ and $C$ factors had

to be adapted due to the very coarse resolution of the data from MPI-ESM or the lack of data on certain parameters of the model. A detailed description of erosion estimation with the adjusted RUSLE model in combination with data from the MPI-ESM model is presented in the supply material. Additionally, due to the overestimation of erosion rates by the adjusted RUSLE model in the Alps, we defined a mean soil erosion rate of $20\,\mathrm{t\,ha^{-1}\,year^{-1}}$ for this region based on high resolution

erosion data from Bosco et al. (2008).
We chose the floodplain deposition fraction to range between 0.5 and 0.8 for natural landscapes that consist out of mainly forest, and between 0.2 and 0.5 for agricultural lands. According to equation 6, $f$ increases exponentially with slope. Based on this we calculated $a_f$ and $b_f$ to be respectively 0.5 and 0.47 for natural landscapes and 0.2 and 0.917 for agricultural land. This means that for low

slopes ($<\pm 0.2\,\%$) in a natural landscape an equal amount of sediment is deposited in the floodplain as on the hillslope, while for agricultural land only $20\,\%$ of the eroded soil from the hillslope will reach the floodplain.
The floodplain residence time is made to range between the median and maximum residence time





of floodplain sediment in the Rhine catchment of respectively 260 and 1500 years. This is in ac-
cordance with the residence times derived from observed sediment storage in the Rhine catchment.
Furthermore, Wittmann and von Blanckenburg (2009) found a residence time of 600 years for flood-
plain sediments at Rees in the Rhine catchment, which falls in the range of the floodplain residence
times of our study. According to equation 3, $\tau$ increases exponentially with flow-accumulation. As
the maximum flow-accumulation is different for different catchments, we used the maximum flow-
accumulation of the Rhine catchment to determine the $a_\tau$ and $b_\tau$ in equation 3. The exact values for
$a_\tau$ and $b_\tau$ are respectively -922442.54 and 165886.77.

### 2.3 Criteria for model evaluation

A large-scale spatial model like the one we presented is difficult to validate due to the lack of large-
scale and long-term observational data. Hoffmann et al. (2013) compiled published data on sediment
storage for regions in Central Europe, mainly for the Rhine catchment, where human induced soil
erosion took place. Combined with a long land use history, where agricultural activities go back
till about 7500 years ago (Houben et al., 2006; Hoffmann et al., 2007), the Rhine catchment
serves as a good case study to investigate the impact of human activities on erosion and sediment yields
through history. The Rhine catchment (Fig. 2) has a size of $\sim 185000\,\mathrm{km}^2$ with a main river channel
length of $\sim 1320$ km and drains large parts of the area between the European Alps and the north
sea. It has a complex topography where the elevation ranges between -180 and 1967 m with a mean
topographical percent slope of 0.07, where percent slopes can go up to 1.2. It consists out of two
large sedimentary catchments (ie, upper Rhine Graben and the lower Rhine Embayment-Southern
North Sea Basin) that serve as large floodplain sinks for sediment, and some upland areas, such as
the Black Forest and the European Alps that serve as major sediment production areas.

From the observed Holocene sediment storage Hoffmann et al. (2013) derived scaling relationships
between storage $S$ ($10^9\,\mathrm{kg} = 1\,\mathrm{Mt}$) and catchment area $A$ ($\mathrm{km}^2$) for floodplains and hillslopes. They
found that for floodplains the sediment storage increases exponentially with catchment area, while
hillslope sediment storage shows a different behavior and increases almost linear with catchment
area. The scaling relationships, given by equation 12 for hillslopes and equation 13 for floodplains,
will be used as the main validation for our sediment budget model.

$$S = (364 \pm 168)10^6 * (\frac{A}{A_{ref}})^{(1.06 \pm 0.07)} \tag{12}$$

$$S = (184 \pm 24)10^6 * (\frac{A}{A_{ref}})^{(1.23 \pm 0.06)} \tag{13}$$

Here, $A_{ref}$ is an arbitrary chosen reference area, in this case $10^3\,\mathrm{km}^2$. The observation data con-
tains 41 hillslope and 36 floodplain sediment storage values, derived from a large number of auger



and bore holes that are used to measure sediment thickness related to human induced soil erosion. Furthermore, Hoffmann et al. (2007) established a Holocene sediment budget for sediments in the floodplains and the delta of the non-Alpine part of the Rhine catchment. They derived sediment thickness of Holocene deposits from borehole data that consists out of 563 drillings and available geological maps. This was then multiplied with floodplain areas to calculate floodplain volumes. Sediments on hillslopes were not addressed in this study. A total floodplain sediment mass of $53.5 \pm 12.4 \times 10^9$ tons was found for the whole Rhine catchment, of which $50\%$ is stored in the Rhine Graben and the delta. The spatial variability of the observed sediment storage will be a second validation test for our model.

Finally, Hoffmann et al. (2007) found an average erosion rate of $0.55 \pm 0.16 \, t \, ha^{-1} \, year^{-1}$ for the last 10000 years, with extreme minimum and maximum values of 0.3 and $2.9 \, \mathrm{t \, ha^{-1} \, year^{-1}}$. However, Hoffmann et al. (2013) included also hillslope sediment storage and calculated a total sediment storage of $126 \pm 41$ Gt for the Rhine catchment, which requires a minimum Holocene erosion rate of approximately $1.2 \pm 0.32 t \, ha^{-1} \, year^{-1}$. This shows that hillslopes are not only the main sources of eroded sediment but can be major millennial-scale sinks for eroded sediment that comes from agriculture. We will compare the erosion rates for the Rhine catchment from our sediment budget model also with the above presented values from the studies of Hoffmann et al. (2007) and Hoffmann et al. (2013).

### 2.4 Simulation setup

In order to simulate sediment storage for a certain catchment, an initial state of that catchment has to be assumed. Here we assume the initial state to be the equilibrium state of a catchment, defined as the state of a catchment where the sediment input is equal to the sediment output, and thus the sediment yield at the outlet of the river should be constant in time. External forces working on a catchment such as land use activities or deglaciation can bring the catchment out of equilibrium in a transient state. In the case of the Rhine catchment the period directly after the Last Glaciation Maximum (LGM) could be of importance due to strong erosion that was triggered by the retreating ice sheets. From today's observations on sediment budgets or erosion rates we cannot determine when the Rhine catchment was in an equilibrium state. Additionally, there are no observations of sediment storage before the start of agricultural activities in the Rhine catchment, which date back to 7500 years ago. This poses a problem in simulating and interpreting the present-day absolute values of sediment storage and yields with our sediment budget model.

In order to still being able to interpret the simulated sediment storage for the Rhine catchment, we will not focus on the absolute values of sediment storage. We will only focus on the change in sediment storage due to land use and climate change since 850AD. Considering mainly the changes induced by external forcing, it is not necessary to know if the system was in an equilibrium or transient state at 850AD. Based on this reasoning, we take the environmental conditions of 850AD



to determine the equilibrium state of the model.

In the rest of this study, we will refer to 850AD as the 'default equilibrium state' that we define based on the mean climate and land cover conditions at 850AD, while one should keep in mind that this

is not the 'real' equilibrium state of the catchment. 850AD is used here as the equilibrium state due to reasons related to data availability, and because human impact in this time period is still small compared to present day.

Hence, our simulation setup structure is generally defined by an equilibrium simulation based on the conditions of 850AD, followed by a transient simulation for the last millennium.

We used climate and land cover data from different simulations of MPI-ESM that were available from the CMIP5 experiment to force the sediment budget model. We performed three equilibrium simulations, one based on the mean climate and land cover conditions of the period 850-950AD, and the two others based on the mean climate and land cover conditions of the mid-Holocene period (6000 years ago) from the mid-Holocene experiment of the MPI-ESM (Table 1). The reason for

performing an equilibrium simulation for the mid-Holocene period is to investigate how different initial conditions for climate and land cover would influence the overall sediment storage change during the last millennium. In the equilibrium simulations the erosion and deposition rates are kept constant and the model is run with a yearly time step till the total floodplain sediment storage of a catchment does not change more than 1 ton per year.

The floodplain and hillslope sediment storage at equilibrium were then used as a starting point for the transient simulation that covers the period 850 - 2005AD. In the transient simulation erosion and deposition rates are averaged over time steps of 100 and 50 years, based on the time resolution of the rainfall erosivity factor ($R$) that is part of the erosion module.

We performed 5 'default' transient simulations, two based on the mid-Holocene equilibrium states,

and three others based on the equilibrium state of the period 850 - 950AD. The different ensemble simulations were used to investigate the uncertainty in the resulting sediment storage due to the input data of MPI-ESM. Additionally, we also performed a climate change and land use change simulation based on the equilibrium state of the period 850 - 950AD (Table 1). In the climate change simulation the land cover was fixed to the mean conditions of the period 850-950AD during the last millennium,

while the climate was variable. In the land use change simulation the climate was fixed to the mean conditions of the period 850-950AD during the last millennium, while the land cover was variable.

## 3   Application of the sediment budget model

### 3.1   Scaling test

In order to validate the sediment budget model we tested if the model can reproduce the scaling rela-

tionships found by Hoffmann et al. (2013) for the non-Alpine part of the Rhine catchment (Eq.12 and 13). For this we chose the grid cells in the Rhine catchment that correspond to the observation points



from Hoffmann et al. (2013). Observation points that fell outside the Rhine catchment, were not considered. When considering only the selected grid cells and applying the same scaling approach as in the study of Hoffmann et al. (2013), we found an average scaling exponent for floodplains of

$1.2 \pm 0.04$ and for hillslopes of $1.05 \pm 0.07$ (Table 2). These values fall in the range of floodplain and hillslope scaling exponents of $1.23 \pm 0.06$ and $1.08 \pm 0.07$ respectively found by Hoffmann et al. (2013). The uncertainty in the scaling exponents is mainly due to the regression, while the uncertainty due to different ensemble simulations is very small (Table 2). Our model also reproduces the characteristic differences in scaling between floodplains and hillslopes as found by Hoffmann et al.

(2013) (Fig. 3a and b). One should note that the grid resolution of the model limits the prediction of sediment storage to grid points with a catchment area $\geq 10^2 \, \mathrm{km}^2$.

When considering all the grid cells of the Rhine catchment we derived a scaling exponent for floodplain storage of $1.33 \pm 0.02$, (Table 3), which is somewhat higher than the value found when only the selected grid cells are used. This may be due to the inclusion of grid cells that lie in the Alpine region

of the Rhine catchment. Including the Alpine region leads then to a stronger gradient in sediment storage and catchment area between the Alps and the Rhine delta. In the Alpine region the model predicts much less sediment storage due to the low residence time and high sediment connectivity, while for the Rhine delta the sediment storage is large due to the high flow-accumulation and high residence times. For hillslope storage the scaling exponent is also slightly higher when including

all grid cells in the scaling approach (Table 3). This can also be explained by including the Alpine region, where the model predicts more sediment storage on hillslopes, compared to the rest of the Rhine catchment due to the high erosion rates in this region.

Furthermore, when including all grid cells in the scaling approach there is more spread in the data, which is clear from the lower r-value of the regression. The small difference between the scaling

exponents when considering all grid cells and the scaling exponents when considering only selected grid cells indicates that the selected observation points from Hoffmann et al. (2013) are robust and representative for the catchment. The relatively small difference can be partly attributed to biases in simulated erosion and deposition rates and the floodplain residence times.

Finally, we found that keeping either the climate or land cover constant throughout the last mil-

lennium had very little impact on the scaling exponent for floodplains. Here, the climate change simulation resulted in a slightly higher and the land use change simulation in a slightly lower scaling exponent. The different forcings had a stronger impact on the scaling for hillslopes, as hillslope sediment storage is only dependent on erosion and deposition rates. For the climate change simulation the scaling exponent for hillslopes increased by $3.8\,\%$, while for the land use change simulation a

small decrease of $0.1\,\%$ was found. This decrease could result from the fact that most land use change took place in the lower parts of the Rhine catchment resulting in an increased sediment storage there. In contrast, the conditions in the Alpine region did not change that rapidly, resulting in a decreased difference in sediment storage on hillslopes between the upper and lower areas in the catchment.





The above results indicate that the scaling relationships are a general feature for the entire Rhine
catchment and are independent of the selected observation points. As the Rhine catchment is a large
catchment with a complex topography, this result indicates that the scaling relationships might be
also applicable for other large river catchments.

**3.2    Origin of scaling between sediment storage and catchment area**

We also performed a sensitivity study to test the robustness of the scaling relationships of the model.
For this we investigated the dependence of the scaling on the three main variables of the model,
namely, the residence time, erosion and deposition. First, we investigated the dependence of the
scaling exponent of floodplains on the residence time. To do this we chose different median resi-
dence times of floodplain sediment in the Rhine catchment, while keeping the maximum residence
time fixed. Changing the median residence time by a factor of 10, from 50 to 500 years, results
in a decrease of $21.8\%$ in the scaling exponent for floodplain storage for the transient simulation
(Table 4). When the median floodplain residence time is increased, the range in the residence time
decreases. This leads to less difference between grid cells with small and large catchment areas in
terms of the sediment loss, and consequently to a decrease in the scaling exponent. We found that
when the residence time is increased by $5.2\%$ (from 50 to 260 years) the scaling exponent decreases
by $18.2\%$, while an increase in the residence time of $1.9\%$ (from 260 to 500 years) results only in
a decrease of the scaling exponent of $4.4\%$. This indicates that the scaling exponent of floodplain
storage does not change linearly with the residence time, and points out that the model shows a
non-linear behavior. The equilibrium simulation shows the same behavior for the scaling exponent
when the residence time is changed. However, here the 10 fold change in the residence time leads to
a slightly larger change in the scaling exponent.

Next, we investigated the dependence of the scaling exponents of floodplains and hillslopes on ero-
sion. We changed the spatial variability of erosion in the Rhine catchment by changing the spatial
variability of the $R$ factor that increased the $R$ values in the Alpine region and decreased the $R$ val-
ues in the rest of the catchment. This resulted in a larger difference between the sediment storage in
small catchment areas and sediment storage in large catchment areas. Although the resulting scaling
exponent for floodplains was still much higher than the scaling exponent for hillslopes, both scaling
exponents increased significantly.

For the deposition we found a minor to neglecting effect on the scaling parameters.

Overall we found that changing erosion and residence time does not change the basic property of
the scaling, which is that floodplain storage grows strongly with catchment area while hillslope
storage shows a linear scaling with catchment area. As the residence time is determined by flow-
accumulation and flow-accumulation determines the spatial variability of floodplain sediment stor-
age, we expect that the scaling parameters of floodplain sediment storage are also mainly determined
by flow-accumulation. Erosion is mainly determined by the slope, and slope determines the spatial



variability of hillslope sediment storage. We expect, therefore, that the slope determines the scaling parameters of hillslope sediment storage. Based on this we argue that the scaling for both flood-plain and hillslope storage is an emergent property of the model and that the scaling parameters are controlled by the underlying topography.

### 3.3 Last millennium sediment storage

We estimated an average soil erosion rate of $2.8 \pm 0.002\,\mathrm{t\,ha^{-1}\,year^{-1}}$ for the last millennium for the entire Rhine catchment. We find that this value is twice as high as the $1.2 \pm 0.32\,\mathrm{t\,ha^{-1}\,year^{-1}}$, which was estimated as the minimum average soil erosion rate for the Holocene by Hoffmann et al. (2013).

The average soil erosion rate for the last millennium resulted in a mean floodplain and hillslope
sediment storage change of $11.95 \pm 0.01$ and $29.68 \pm 0.03$ Gt, respectively, for the last millennium (Table 5). Altogether, floodplain and hillslope storage result in $41.63 \pm 0.02$ Gt of sediment, which can be considered as the contribution of climate and land use change to sediment storage in the last millennium. It is, however, hard to say what the range in the change of sediment storage should be for this period, as there are no related studies for this specific time period. Hoffmann et al. (2007)
found a total sediment storage of $126 \pm 41$ Gt for the Holocene in the Rhine catchment. Our values are lower than this range found by Hoffmann et al. (2007), due to the fact that we only consider the impact of last millennium on the sediment storage and not the last 7500 years. Our results show that the sediment storage of the last millennium form 25 to $50\%$ of the total sediment storage of the last 7500 years. This indicates that the average sediment storage rate during the last millennium
is higher than the average rate during the last 7500 years. This supports the findings from previous studies (Bork, 1989; Notebaert et al., 2011), which show that land use change has a significant and long-term impact on erosion and sediment mobilization.

Furthermore, Hoffmann et al. (2013) found a floodplain to hillslope ratio of about 0.88, indicating that during the Holocene more sediment was stored on hillslopes than in floodplains. We find with
our model a floodplain to hillslope ratio of about 0.46, confirming that more sediment is stored on hillslopes. However, the floodplain to hillslope ratio from our model indicates a larger difference in sediment storage between floodplains and hillslopes than in the study of Hoffmann et al. (2013). This can be attributed to the simple representation of the sediment deposition process in our model, and may indicate that a more complex representation of deposition is needed where for example the
effect of the roughness of the landscape is explicitly included.

We also analyzed the spatial variability of the modelled sediment storage, and found that the model reproduces the spatial variability well when compared to the observed values from Hoffmann et al. (2007) for the Holocene (Fig. 4). Here we found a correlation coefficient of 0.76, where sediment storage in floodplains increased with the catchment area. Furthermore, we found that most flood-
plain sediment is stored in the Mosel sub-catchment, in contrast to the observations that show that





most of the sediment is stored in the Upper-Rhine sub-catchment (Table 6). This can be related to the fact that different dynamical processes play a role in the Upper-Rhine catchment, which are triggered by the Alps. Melting ice sheets for example can produce a lot of erosion that is not captured by our model and in this way the total stored sediment in the catchment could be underestimated.

Furthermore, the Mosel sub-catchment has a highly complex topography, which may indicate that our sediment budget model is too coarse for an accurate representation of floodplain storage for this catchment.

For hillslope sediment storage we found a similar spatial trend as for the floodplain sediment storage, with some more variation between the minimum and maximum values (Table 6). Also here, the

Model catchment has the most sediment stored. Furthermore, when comparing floodplain to hillslope sediment storage we find that the floodplain to hillslope ratio varies significantly between the various sub-catchments. The highest ratio of 0.48 is found for the Lower Rhine sub-catchment, while the lowest ratio of 0.14 is found for the Emscher sub-catchment. The ratios seem not to be correlated with slope or catchment area and can be assumed as independent features of the model.

The sediment budget model presented here, has been developed to simulate long-term historical trends and to determine the main drivers behind these trends. Figure 5 shows the land use change and the 10 year-mean precipitation timeseries averaged over the Rhine catchment for the last millennium. There are two interesting periods, respectively, 1350-1400AD and 1750-1950AD that show increased precipitation amounts correlating with a sudden increase in land use change (increase in

crop and pasture). Both periods lead to maxima in the erosion timeseries of $2.8\,\mathrm{t}\,\mathrm{ha}^{-1}\,\mathrm{year}^{-1}$ and $4.3\,\mathrm{t}\,\mathrm{ha}^{-1}\,\mathrm{year}^{-1}$, respectively (Fig. 6a and 6b). This corresponds to increased erosion rates during the 14th and 18th century found by (Bork, 1989; Lang et al., 2003) for Germany.

We find the strongest increase in the sediment storage rate for floodplains during the period 1750-1850AD, while for hillslopes during the period 1850-1950AD. For hillslopes this maximum sedi-

ment storage rate corresponds to a maximum increase in the deposition rate, which is a result of a maximum increase in land use change and a high erosion rate.

Land use change leads to a sediment disconnectivity in the landscape, which prevents the sediment stored on hillslopes of reaching the fluvial system on the timescale of the last millennium. In contrast to hillslopes, the maximum sediment storage rate for floodplains is a result of the interplay

between deposition and sediment loss from the catchment. In the period 1750-1850AD land use change started to increase in the Alpine region, which did not experience such a strong change in land-use as the lower parts of the catchment before this time period. During this period, the deposition to floodplains increased significantly due to the increased erosion rates as a result of land use change. Also, land use change started to impact the Alpine region, where areas with steep slopes

and short residence times lead to a strong sediment flux downstream. However, due to the long residence time of the areas located downstream, the sediment loss from the catchment did not increase as much, leading to an increased sediment storage in the floodplains. This is in accordance with




the findings of Asselman et al. (2003), who found that due to an inefficient sediment delivery, an increase in soil erosion in the Alps will have a little effect on sediment load downstream the Rhine catchment.

Furthermore, if we disentangle the effects of land use and climate on the sediment storage for floodplains and hillslopes, we can see that land use change explains most of the change in sediment storage. For floodplains climate change also has a non-negligible impact on the temporal variability of sediment storage. For example in the periods 1350-1400AD and 1750-1950AD, the sediment storage rate is increased due to increased precipitation that leads to a strong sediment flux downstream from upstream areas. If the land use change conditions of the period 850 and 950AD were kept constant, the total change in sediment storage in floodplains and hillslopes during the last millennium would be 2.9 and 15.4 Gt, respectively. This is four and two times, respectively, less than the change in floodplain and hillslope sediment storage when land use change is variable (Fig. 7a and 7b). When the land cover is kept constant, the overall sediment storage still increases for the climate change scenario due to the overall increased trend in precipitation during the last millennium. If only the climate change conditions are kept constant, the resulting change in sediment storage in floodplains and hillslopes would be 10 and 27.4 Gt, respectively.

### 3.4 Uncertainty assessment and limitations of the modelling approach

As shown in the previous sections, the average erosion rate for the Rhine catchment is found to be overestimated when compared to the erosion rate for the Holocene from the study of Hoffmann et al. (2013). As we consider in this study only the last millennium, where human impacts through land use change are strongest pronounced, it is logical that our estimated average soil erosion rate is higher. For present day, we found an average soil erosion rate of $3.3\,\mathrm{t\,ha^{-1}\,year^{-1}}$ for the non-Alpine part of the Rhine catchment, which is also overestimated when compared to other studies. Cerdan et al. (2010) found for the non-Alpine part of the Rhine catchment a value of $1.5\,\mathrm{t\,ha^{-1}\,year^{-1}}$, while Auerswald et al. (2009) found for Germany a value of $2.7\,\mathrm{t\,ha^{-1}\,year^{-1}}$.

Comparing the spatial variability of erosion rates for present day with the high resolution estimates from Cerdan et al. (2010), we find that erosion is overestimated for the whole Rhine catchment. We expect that the overestimation in the modelled erosion rates is mainly due to uncertainties related to the coarse input datasets on climate and land cover, and biases in the adjusted RUSLE model.

We found that precipitation is overestimated by MPI-ESM for the Rhine catchment. Even after introducing a correction factor, which partly adjusted the $R$ value estimation to values from observational datasets, biases related to the $R$ factor remain.

Additionally, coarse resolution land cover fractions and leaf area index ($LAI$) from MPI-ESM also affect the total erosion rates. Using coarse resolution data to calculate the $C$ factor of the adjusted RUSLE model results in discrepancies between the $C$ and $S$ factors. For example, consider a large grid cell with a complex topography where cropland is allocated in flat areas and forest in the steeper



areas. Even though the $C$ factor is calculated correctly as combination of cropland and forest frac-
tions, it is applied to the whole grid cell. This leads to an overestimation of erosion rates for flat
areas, as erosion is in the first order controlled by the slope through the $S$ factor. We attempted to
correct this by introducing slope classes for each coarse grid cell with resolution of MPI-ESM (1.875
degrees). The cropland was then allocated to the flatter areas, while in the steeper areas more of the
other land cover types was allocated. However, this only had a minor effect on the overall erosion
rates, indicating that this is not the major source for the overestimated erosion rates.

Additionally, the absence of the seasonality in the $C$ factor results in discrepancies between the $C$
and $R$ factors.

Neglecting the support practice ($P$) and slope-length ($L$) factors in the adjusted RUSLE model also
affect the erosion rates. As the Rhine catchment has a long land use history, land management strate-
gies were implemented historically, to decrease soil erosion rates. We partly captured the effects of
land management in the $C$ factor, however, we expect that introducing the $P$ and $L$ factors in the
model will reduce the soil erosion rates in cropland.

Also, biases in the adjusted RUSLE model, such as the unadjusted $C$ and $K$ factors and the low per-
formance of the model in mountainous areas, have an equally important effect on the total erosion
rates.

Another large uncertainty in our sediment budget model, besides the biases in erosion rates, is the
choice of the equilibrium state. We found a decreasing trend in the floodplain sediment storage in
the transient simulation when using the equilibrium state based on the mean conditions of 6000 BP.
This can be attributed to the different spatial distribution of erosion and the average high erosion
rate for the mid-Holocene of $7.8\,\mathrm{t\,ha^{-1}\,year^{-1}}$. When switching from the equilibrium state to the
transient state, the erosion rates drop and the spatial distribution changes significantly. This leads to
a decreased sediment flux from upstream areas and overall decreased sediment production rates that
result in a drop in sediment storage in the floodplains. For the hillslopes we found that the equilib-
rium state has minimal to no influence on the total sediment storage for the last millennium.

The initial conditions determine the amount and spatial distribution of erosion in the catchment dur-
ing the time that the model runs to equilibrium. Therefore, the equilibrium state that is then reached,
largely determines the spatial distribution, trend, and amount of the sediment storage during the tran-
sient period.

Finally, the different ensemble simulations for the period 1850-2005AD do not differ strongly in pre-
cipitation and land cover/land use change, and therefore do not contribute much to the uncertainty in
the overall erosion rates and sediment storage. This period is also too short to find significant effects
on the sediment storage from different ensemble simulations.

There are also some limitations to the model. The sediment yield cannot be reproduced for catch-
ments where the initial state of the catchment is uncertain. However, with accurate data input on
climate and land cover, the model can be made applicable for tropical catchments on the timescale



of the last millennium, after adjusting the model parameters for these catchments. This is because
we expect the effect of the last glaciation to be minimal on tropical catchments. In combination with
low human activities in 850AD assuming an equilibrium state for these catchments in 850AD seems
reasonable. This can be tested in the future by applying the model on other large catchments.

Furthermore, a more concrete parameterization for the residence time and deposition of floodplain
sediment, and a possible new parameterization for the residence time of hillslope sediment could
lead to an improvement of the model. Finally, more validation with long-term sediment storage from
other catchments, especially tropical catchments, would be an important contribution in making the
model better applicable on a global scale.

**4  Conclusions**

In this study we introduced a new model to simulate long-term, large-scale soil redistribution based
on the sediment mass-balance approach. The main objective here was to develop a sediment budget
model that is compatible with Earth System Models (ESMs), to simulate large-scale spatial patterns
of soil erosion and redistribution for floodplains and hillslopes following climate change and land

use change. We applied this sediment budget model on the Rhine catchment as a first attempt to
investigate its behavior and validated the model with observed data on sediment storage and erosion
rates.

We show that the model reproduces the scaling relationships between catchment area and sediment
storage found in observed data from Hoffmann et al. (2013). These scaling relationships show that

the floodplain storage increases significantly with catchment area while the hillslope storage scales
linearly with catchment area. The scaling exponents can be modified by changing the spatial distri-
bution of erosion or by changing the residence time for floodplains. However, the main feature of
the scaling relationships, which is that floodplain storage increases stronger with catchment area as
hillslopes, is not changed. Based on this we conclude that the scaling relationships are an emergent

feature of the model and mainly dependent on the underlying topography.

We found a mean soil erosion rate of $2.8 \pm 0.002\,\mathrm{t\,ha^{-1}\,year^{-1}}$ for the last millennium (850 -
2005AD). This is an overestimation when compared to the minimum Holocene erosion rate of
$1.2 \pm 0.32\,\mathrm{t\,ha^{-1}\,year^{-1}}$ from Hoffmann et al. (2013). Also for present day the erosion rates from
our model are overestimated. We argue that this is mainly due to the coarse resolution input data

on climate and land cover, and the fact that the land cover factor of the erosion model is not ad-
justed for a coarse resolution application. Additionally, the absence of the seasonality in the $C$ factor
plays a role, and other biases of the adjusted RUSLE model, such as the neglection of the land man-
agement and slope-length factors. However, we aim with the sediment budget model to distinguish
between the floodplain and hillslope sediment storage, simulate their long-term behavior, and more

specifically estimate the spatial distributions rather than the total amounts. For this objective a coarse



estimation of erosion is sufficient.

The simulated erosion rates resulted in a change in floodplain and hillslope sediment storage during the last millennium of $11.95\pm0.03$ and $29.68\pm0.01$ Gt, respectively. Based on this and the observed data we estimate that the climate and land use changes during the last millennium contribute between

25 - 50% to the total sediment storage for the past 7500 years.

Disentangling the contribution from climate change and land use change on the change in sediment storage during the last millennium for the Rhine catchment, we find that the climate change simulation results in a total change in sediment storage in floodplains and hillslopes of 2.9 and 15.4 Gt, respectively. While the land use change simulation results in a total change in sediment storage

in floodplains and hillslopes of 10 and 27.4 Gt, respectively. This shows that land use change contributes most to the change in sediment storage during the last millennium for the Rhine catchment. Furthermore, the model reproduces the overall spatial distribution of floodplain sediment storage of the last millennium. However, there are some outliers, such as the Mosel catchment for which the model simulates a too high sediment storage. This could be a result of biases in the erosion rates and

the fact that our model is limited to the last millennium. We also found that the hillslope storages of the sub-catchments show a similar spatial pattern as the floodplain storage.

When analyzing the timeseries of erosion rates during the last millennium we found that the model reproduces the timing of the maxima in erosion rates as found in the study of Bork (1989). We also find that land use change is the main driver behind the trends in erosion and sediment storage for

both floodplains and hillslopes. For floodplains, however, climate change has a non-negligible impact on the temporal variability of sediment storage. When keeping the land cover constant to the conditions in the period 850 to 950AD, we find that the sediment storage still increases due to an increased trend in precipitation in the last millennium.

We conclude that our sediment budget model is a promising tool for estimating large-scale long-term

sediment redistribution. An advantage of this model is its capability to use the framework of ESMs to predict trends in sediment storage and yields for the past, present and future.

*Acknowledgements.* The article processing charges for this open-access publication have been covered by the Max Planck Society. J.Pongratz was supported by the German Research Foundation's Emmy Noether Program (PO 1751/1-1).



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



**Figure 1a.**

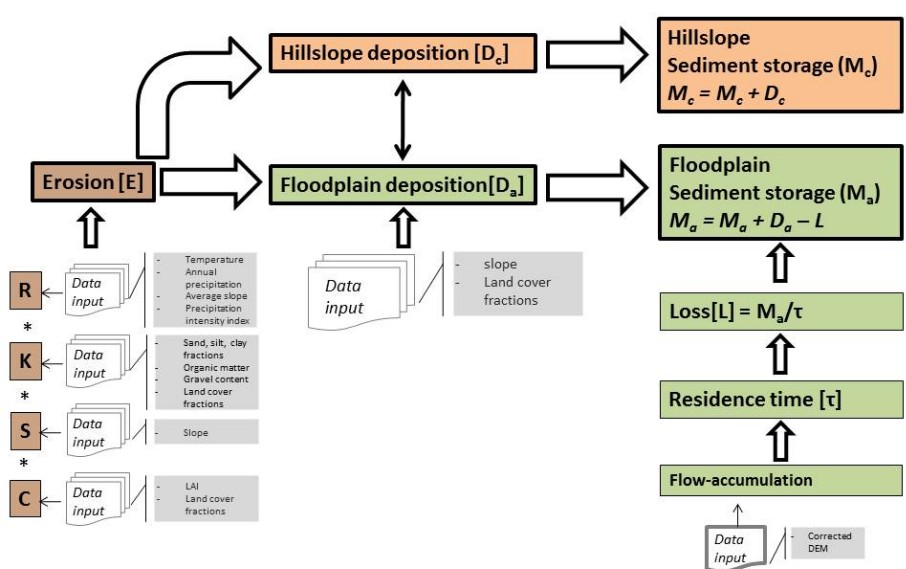

**Figure 1b.**

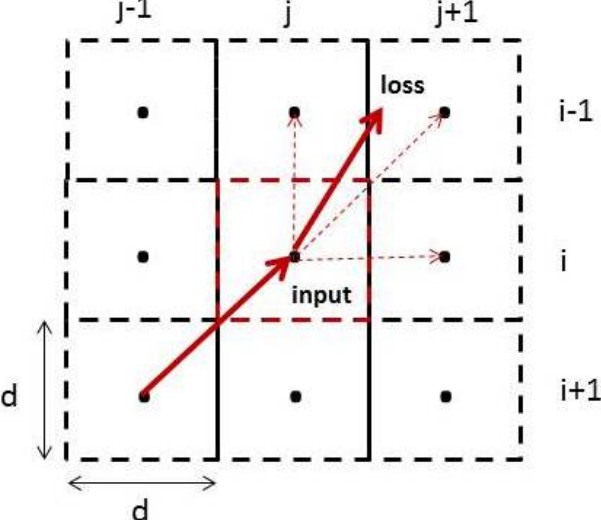

**Figure 1.** Model scheme (a) with multiple flow routing (b)





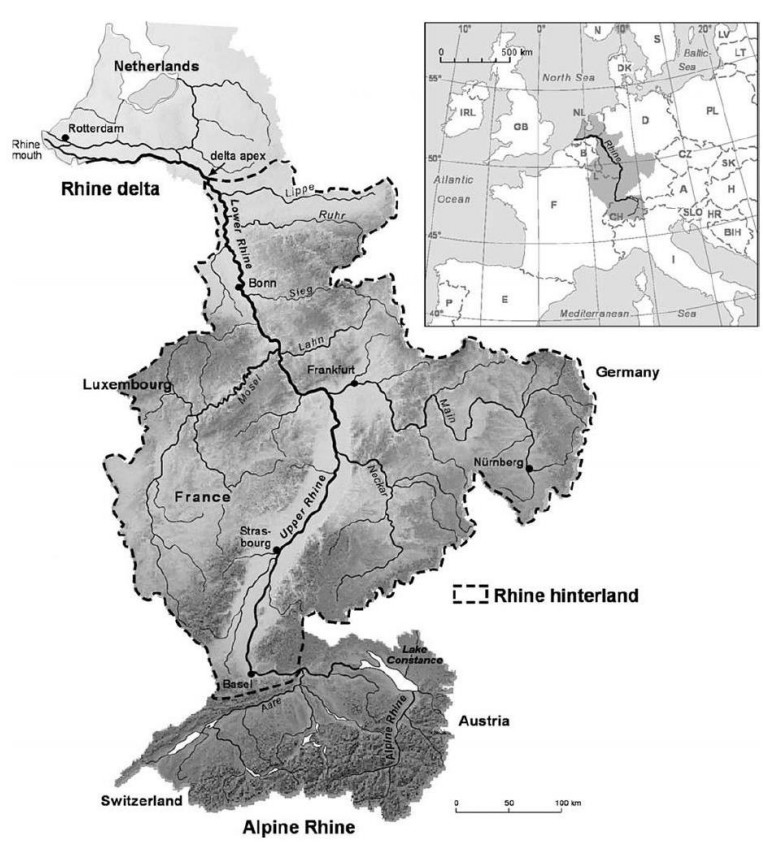

**Figure 2.** The Rhine catchment (Hoffmann et al., 2013)



**Figure 3a.**

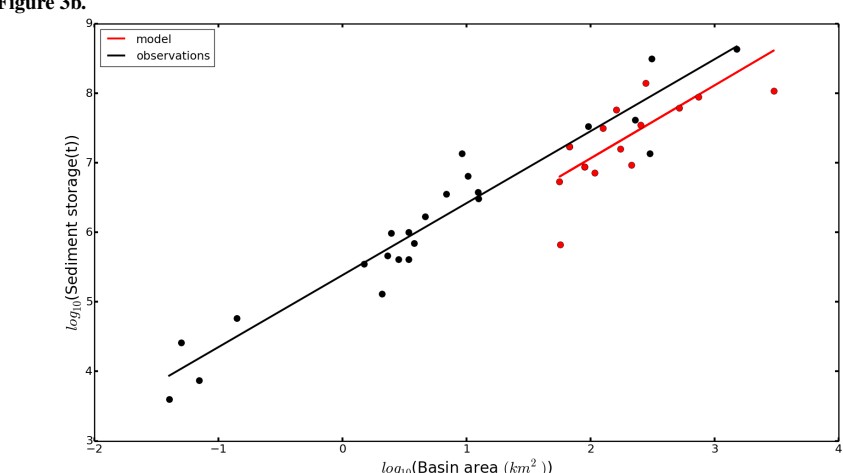

**Figure 3b.**

**Figure 3.** Scaling of floodplain (a) and hillslope (b) sediment storage from the transient simulation in the non-Alpine part of the Rhine catchment. The black dots and black trend line correspond to the observed sediment storage values from Hoffmann et al. (2013). The colored dots and colored trend line correspond to modelled sediment storage values that correspond to the observation points from Hoffmann et al. (2013) and fall into the borders of the Rhine catchment.




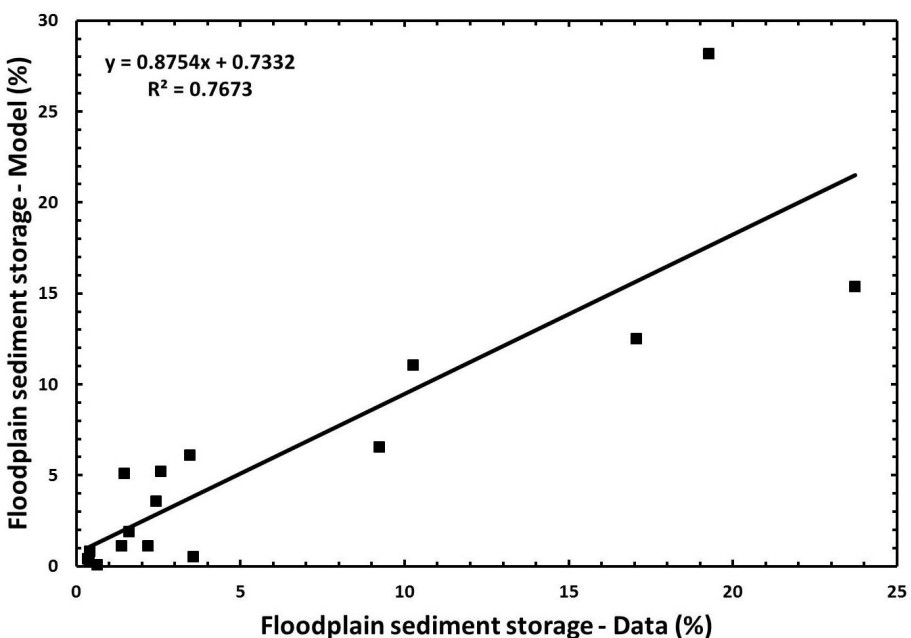

**Figure 4.** Observed versus modelled floodplain sediment storage for Rhine sub-catchments. The values are in percentage (actual storage divided by the sum times 100). Data on the observed sediment storage is taken from Hoffmann et al. (2007).



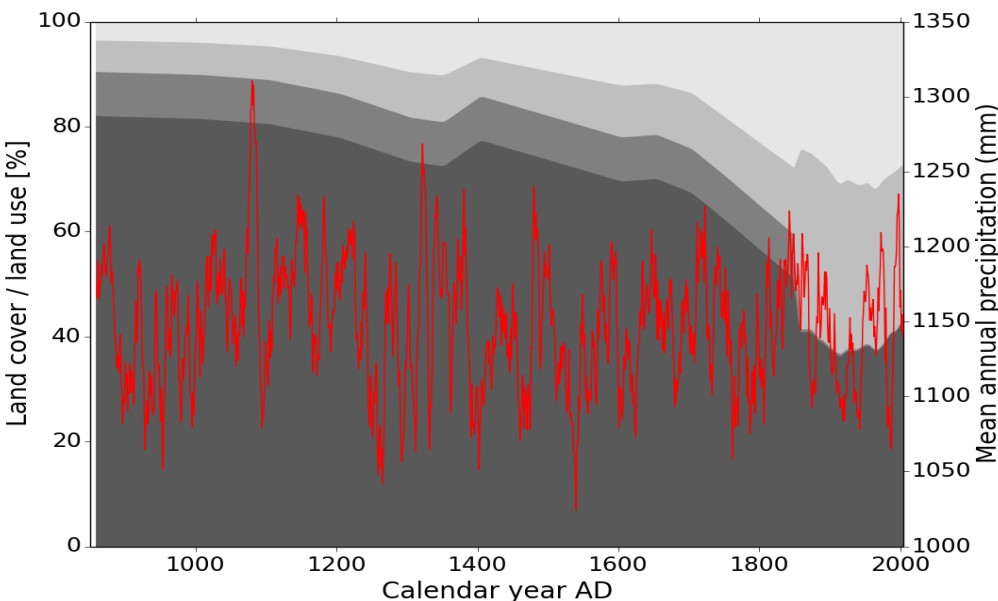

**Figure 5.** Land cover and precipitation variability averaged over the Rhine catchment for the last millennium. The red line is the 10 year mean total precipitation for the Rhine catchment. The background colors are land cover types, starting from the darkest grey to the lightest: forest, bare soil, grass, crop and pasture. Land cover and precipitation data is from MPI-ESM.





**Figure 6a.**

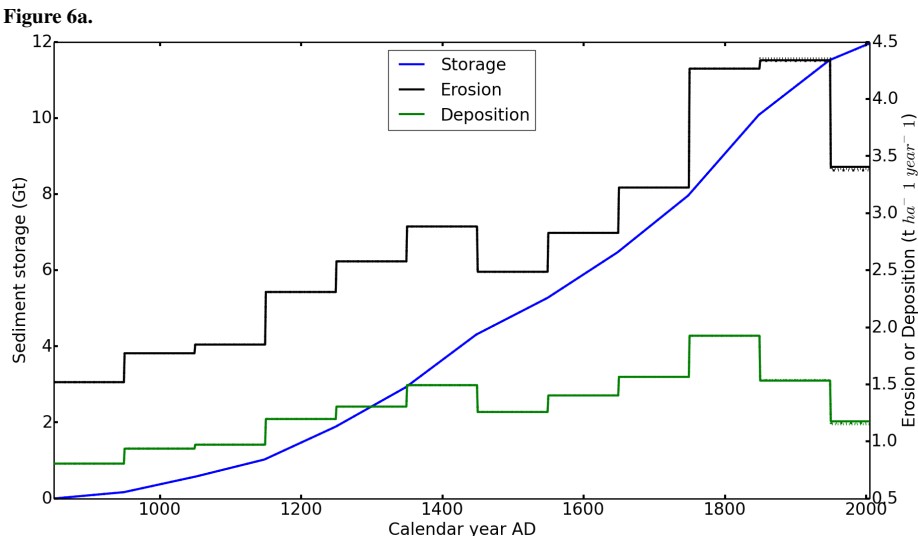

**Figure 6b.**

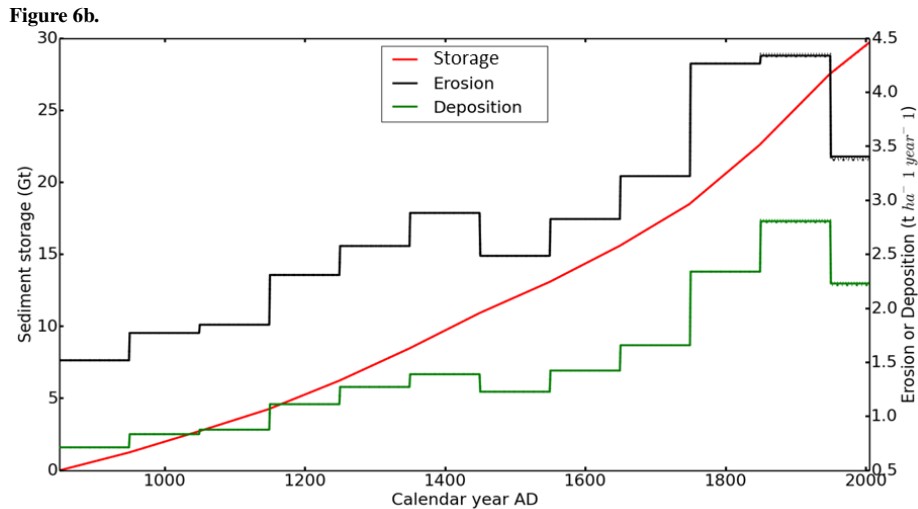

**Figure 6.** (a) Timeseries of simulated average erosion (black line), average deposition (green line) and the total change in sediment storage (blue line) with respect to 850AD for floodplains in the last millennium in the Rhine catchment. (b) Timeseries of simulated average erosion (black line), average deposition (green line) and the total change in sediment storage (blue line) with respect to 850AD for hillslopes in the last millennium in the Rhine catchment.




**Figure 7a.**

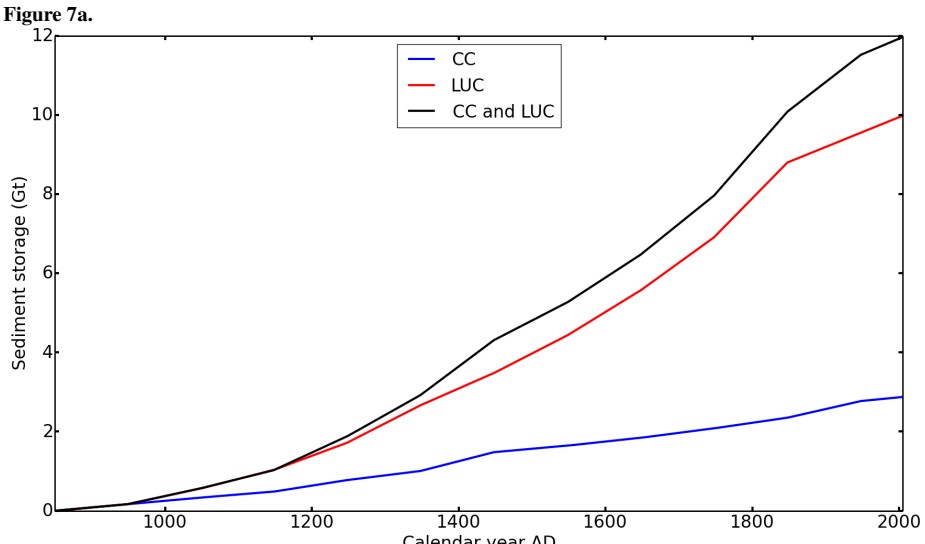

**Figure 7b.**

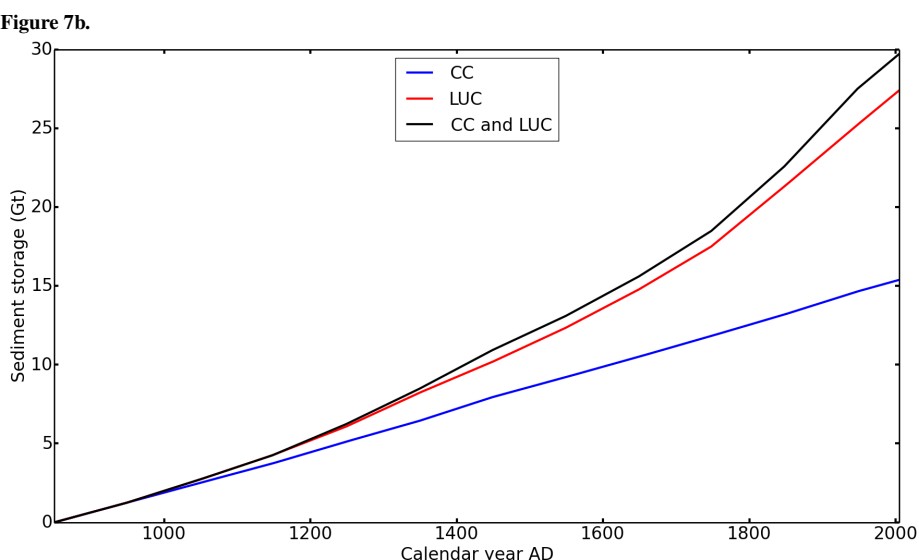

**Figure 7.** Simulated change in (a) floodplain and (b) hillslope sediment storage for the Rhine catchment during the last millennium. Shown are the sediment storage for the climate change simulation, where land cover is set to the conditions of the period 850-950AD (CC - blue line), the sediment storage for the land use change simulation, where the climate is set to the conditions of the period 850-950AD (LUC - red line), and the sediment storage where both climate and land cover change during the last millennium (CC and LUC - black line).



**Table 1.** Simulation specifications for the application of the sediment budget model for the Rhine catchment. For each experiment with the sediment budget model the type of simulation (equilibrium or transient), the time period, and the initial conditions on which the simulation is based on, are given. Furthermore, we also provide the number of simulations we made with the model for a certain type of simulation, and the experiment from MPI-ESM that we used to derive the input data to force the sediment budget model.

| Experiment | Simulation | Time period | Initial conditions | Experiment MPI-ESM | number of ensemble simulations |
|---|---|---|---|---|---|
| | equilibrium | | 850-950AD | last millennium | 1 |
| | equilibrium | | 6000 BP | mid-Holocene | 2 |
| default | transient-part1 | 850-1850AD | 850-950AD | last millennium | 1 |
| default | transient-part2 | 1850-2005AD | transient-part1 | historical | 3 |
| default | transient-part1 | 850-1850AD | 6000 BP | last millennium | 2 |
| default | transient-part2 | 1850-2005AD | transient-part1 | historical | 2 |
| climate change | transient-part1 | 850-1850AD | 850-950AD | last millennium | 1 |
| climate change | transient-part2 | 1850-2005AD | transient-part1 | historical | 1 |
| land use change | transient-part1 | 850-1850AD | 850-950AD | last millennium | 1 |
| land use change | transient-part2 | 1850-2005AD | transient-part1 | historical | 1 |

**Table 2.** Summary of regression results of sediment storage scaling at the end of the equilibrium and transient simulations. Here we consider only the grid cells that correspond to the observation points from Hoffmann et al. (2013) and fall into the borders of the Rhine catchment. The r-value is the Pearson correlation coefficient, and the slope and intercept are the scaling parameters.

| | Floodplains | | | Hillslopes | | |
|---|---|---|---|---|---|---|
| | slope | intercept | r-value | slope | intercept | r-value |
| Equilibrium | $1.659 \pm 0.037$ | $3.123 \pm 0.130$ | 0.99 | $1.085 \pm 0.060$ | $6.429 \pm 0.180$ | 0.94 |
| Transient ensemble 1 | $1.198 \pm 0.038$ | $3.877 \pm 0.133$ | 0.98 | $1.050 \pm 0.064$ | $4.963 \pm 0.193$ | 0.93 |
| Transient ensemble 2 | $1.202 \pm 0.038$ | $3.853 \pm 0.133$ | 0.98 | $1.048 \pm 0.065$ | $4.971 \pm 0.194$ | 0.93 |
| Transient ensemble 3 | $1.203 \pm 0.038$ | $3.85 \pm 0.133$ | 0.98 | $1.048 \pm 0.065$ | $4.972 \pm 0.194$ | 0.93 |
| Hoffmann et al. (2013) | $1.230 \pm 0.060$ | 4.450 | 0.96 | $1.080 \pm 0.070$ | 5.380 | 0.96 |

**Table 3.** Summary of regression results of sediment storage scaling after the equilibrium and transient simulations. Here we consider all grid cells in the Rhine catchment area. The r-value is the Pearson correlation coefficient, and the slope and intercept are the scaling parameters.

| | Floodplains | | | Hillslopes | | |
|---|---|---|---|---|---|---|
| | slope | intercept | r-value | slope | intercept | r-value |
| Equilibrium | $1.685 \pm 0.015$ | $2.827 \pm 0.039$ | 0.80 | $1.118 \pm 0.016$ | $6.327 \pm 0.040$ | 0.62 |
| Transient ensemble 1 | $1.330 \pm 0.017$ | $3.406 \pm 0.042$ | 0.67 | $1.111 \pm 0.015$ | $4.741 \pm 0.039$ | 0.63 |
| Transient ensemble 2 | $1.332 \pm 0.017$ | $3.401 \pm 0.042$ | 0.67 | $1.112 \pm 0.015$ | $4.740 \pm 0.039$ | 0.63 |
| Transient ensemble 3 | $1.332 \pm 0.017$ | $3.400 \pm 0.042$ | 0.67 | $1.112 \pm 0.015$ | $4.741 \pm 0.039$ | 0.63 |





**Table 4.** Summary of regression results of the sensitivity analysis on floodplain sediment storage scaling. Here we consider only the previously mentioned selected grid cells in the Rhine catchment area. The r-value is the Pearson correlation coefficient, and the slope and intercept are the scaling parameters.

|  | slope | intercept | r-value |
|---|---|---|---|
| *Equilibrium* | | | |
| $\tau$ median = 50 years | $1.787 \pm 0.041$ | $2.143 \pm 0.143$ | 0.99 |
| $\tau$ median = 260 years | $1.659 \pm 0.037$ | $3.123 \pm 0.13$ | 0.99 |
| $\tau$ median = 500 years | $1.616 \pm 0.037$ | $3.496 \pm 0.128$ | 0.99 |
| *Transient* | | | |
| $\tau$ median = 50 years | $1.464 \pm 0.055$ | $2.59 \pm 0.193$ | 0.97 |
| $\tau$ median = 260 years | $1.198 \pm 0.038$ | $3.877 \pm 0.133$ | 0.98 |
| $\tau$ median = 500 years | $1.145 \pm 0.035$ | $4.128 \pm 0.122$ | 0.98 |

**Table 5.** Summary of sediment storages $M$ (Gt), erosion ($E$) and deposition ($D$) rates in $t\,ha^{-1}\,year^{-1}$, and the related uncertainty ranges for the Rhine catchment for the period 850-2005AD. The uncertainty values represent the range in the mean values due to different ensemble simulations.

|  | Mean $M$ | Ensemble uncertainty $M$ | Mean $E$ | Ensemble uncertainty $E$ | Mean $D$ | Ensemble uncertainty $D$ |
|---|---|---|---|---|---|---|
| Floodplains | 11.95 | 0.01 | 2.787 | 0.0015 | 1.296 | 0.0005 |
| Hillslopes | 29.68 | 0.03 | 2.787 | 0.0015 | 1.491 | 0.0015 |
| Whole Rhine catchment | 41.63 | 0.02 | 2.787 | 0.0015 | 2.787 | 0.0015 |

**Table 6.** Observed versus modelled sediment storage (Gt) for Rhine sub-catchments. The catchment area is given in $km^2$. Data on the observed sediment storage is taken from Hoffmann et al. (2007).

| Catchment | Catchment area | Observed floodplain storage | Modelled floodplain storage | Modelled hillslope storage |
|---|---|---|---|---|
| Lippe | 4858 | 1.62 | 0.03 | 0.07 |
| Lower Rhine | 404 | 0.99 | 0.07 | 0.14 |
| Emscher | 806 | 0.29 | 0.005 | 0.03 |
| Ruhr | 4477 | 1.10 | 0.21 | 0.68 |
| Wupper | 838 | 0.18 | 0.02 | 0.06 |
| Erft | 1819 | 0.63 | 0.07 | 0.22 |
| Sieg | 2870 | 0.73 | 0.11 | 0.38 |
| Lahn | 5916 | 1.57 | 0.36 | 1.15 |
| Wied | 745 | 0.16 | 0.02 | 0.13 |
| Ahr | 911 | 0.19 | 0.05 | 0.15 |
| Middle Rhine | 1046 | 0.66 | 0.30 | 0.87 |
| Main | 27307 | 7.75 | 0.73 | 2.66 |
| Mosel | 28227 | 8.75 | 1.64 | 4.93 |
| Nahe | 4070 | 1.17 | 0.30 | 1.11 |
| Upper Rhine | 3006 | 10.77 | 0.90 | 2.69 |
| Neckar | 13971 | 4.19 | 0.38 | 1.93 |
| Ill | 4858 | 4.66 | 0.65 | 2.28 |