# Peer review of "Modelling long-term, large-scale sediment storage using a simple sediment budget approach"

_Earth Surface Dynamics, 2015_

## Referee Comment (RC1) · A. Chappell (Referee) · 15 Feb 2016

Review overall

I think the manuscript describes well the rationale for the emerging priorty of representing soil redistribution in Earth System Models. The authors have set out a clear model and its assumptions and then described some of the approximations necessary to parameterise the model. On that basis the model is used to make estimates of erosion and deposition during the last millenium. The results are compared to estimates made by Hoffman and colleagues. Usefully, the authors provide an assessment of uncertainty in their modelling. Overall, I think the manuscript will make a useful contribution to the literature and could be published with revision. I provide below comments which support my conclusion and which I hope can be useful to the authors in their revision.
[Figure]

Comments and points for clarification

1) I think equation 10 is a description of net soil redistribution by water. The E provides gross water erosion and 1-f provides an adjustment based on the "remaining soil that has not been transferred to the floodplain directly". I think this approximation of net soil redistribution by water should be made clear as I think this emphasises the need for and significance of a net approach. I also think that (if I have understood the approach correctly) you should be able to validate your approximation against maps of net soil redistribution e.g., Australia. Note that 137Cs-derived maps for net soil redistribution across Australia include wind and water erosion and to validate only the water component the gross wind erosion would need to be removed (for which data is also available for Australia).

2) I think I understand why you needed to choose a grid resolution for your study. However, if the modelling is to be used in global ESM then it will need to be independent of grid resolution. I suspect that its application to other larger and flatter catchments would be represented by much of your model but perhaps not in the assumption that 5 arcmin is optimal for representing floodplain and hillslope. For example, Australia has some very large catchments which could mean that your chosen grid resolution may have no floodplain. I agree that a finer resolution would also not work in many regions but would work in other very steep, rugged terrain regions.

3) There is very little justification for the model parameter values in section 2.2 (page 8). You may be interested in considering for this manuscript and future work whether your approximation for f is consistent with the Multi-resolution Valley Bottom Flatness (MrVBF) data for Australia can be found on the link below.

https://data.csiro.au/dap/landingpage?pid=csiro%3A5681

The land cover type and metrics are readily available Australia and presumably for precent slope so that might provide additional calibration / validation for your model. I'm not convinced about the use of average slope (as opposed to median) and which

will be influenced by the resolution of the grid.

4) I think if scaling is to be an important part of the paper, as demonstrated by section 3.1 and abstract, then some background on Hoffman et al. (2013) needs to be included so that this topic in the manuscript is easier to comprehend.

5) I think it reasonable to qualify the extent to which this modelling can be used generally within ESMs to model soil redistribution by making explicit the contribution of soil redistribution by wind. In many global regions and drier more gusty phases in the past (and potentially the future), wind erosion may be considerably more important than water erosion. In other semi-arid regions the interplay between downslope erosion by water and removal by wind may cause a net soil redistribution to tend towards little change for some long period. In any case, I think it may be worthwhile for modellers interested in implementing your model that the wind erosion and dust emission component is important for soil redistribution. I accept that these processes may not be particularly relevant in the Rhine but note that they are in the sandy previous outwash plains of other parts of Germany and NW Europe.

6) I think the grammar and syntax of the manuscript need to be improved e.g., focus on consistent tenses. This might also be a good time to reduce the duplication of background material in the Introduction cf. start of Methods section. I think much of the technical justification might be moved from the Introduction to the start of the Methods section.

7) References need some attention e.g., Oost or Van Oost

**ESurfD**

---

## Referee Comment (RC2) · B. Guenet (Referee) · 16 Feb 2016

The paper by Naipal et al is a quite interesting study on the implementation of erosion schemes to Earth system models. This an hot topic and this study is a significant contribution to existing literature. In particular, it provides a very flexible approach that can be adapted to any ESM. I believe that after some revisions the paper can be published. I have some comments detailed below that I hope will be helpful to improve the paper.

1. You used a modified version of the RUSLE equations without the support practice factor and then conclude that land use change is a driving factor of erosion. I think that you should discuss carefully how their conclusions would be impacted by the use of the support practice factor. In particular, how it could change the trends after the 1950's

[Figure]

and during the middle age when animal traction was more and more used to plough.

2. You used a modified version of RUSLE on non-croplands areas whereas this equation has been developed on croplands areas. I missed few sentences to justify that the use of RUSLE makes sense also for forest and grassland.

3. It would have been quite interesting to know the sensitivity of your approach to the inputs coming from the MPI-ESM using simulations coming from other ESMs. I am aware that this is asking a lot of additional work to redo everything using other ESMs outputs therefore adding just few elements in the discussion will be enough but at least it is important to mention it and to discuss how the uncertainties from the ESMs results might impact your conclusions.

4. The discussion refers several times to the land use history but without enough references. Please document better this part.

Minor comments: L 133: You implicitly assume that movement of water during the flooding events do not induce erosion? If I understood well it should be clearly stated.

L 151: I am not sure to fully understand what at and bt mean physically. Please clarify.

L 355: Does it means that you use the same climate each year without inter-annual variability or do you repeat the sequence between 850 and 950 AD?

Fig. 4: Since it is scatter plot, you should fix the intercept to zero to have a better idea on how close to the 1:1 line the model is.

Supplementary material l 44: If I understood well these parameters are fixed during the simulations? Why not use the stock of organic C predicted by the MPI ESM?
* * *

---

## Author Comment (AC1) · 21 Mar 2016

We would like to thank Adrian Chappell for his constructive comments. In this response we provide an answer to all the comments and the indicated changes will be applied in the revised manuscript.

Comment 1: "I think equation 10 is a description of net soil redistribution by water. The E provides gross water erosion and 1-f provides an adjustment based on the "remaining soil that has not been transferred to the floodplain directly". I think this approximation of net soil redistribution by water should be made clear as I think this emphasises the need for and significance of a net approach. I also think that (if I have understood the approach correctly) you should be able to validate your approximation against maps of net soil redistribution e.g., Australia. Note that 137Cs-derived maps for net soil

redistribution across Australia include wind and water erosion and to validate only the water component the gross wind erosion would need to be removed (for which data is also available for Australia)"

Answer part 1: Our approach emphasizes the importance of simulating each of the main processes related to sediment dynamics by water, which are erosion, deposition and transport. As such, we do not only study the net export of sediment from a catchment but also explicitly represent both net and gross erosion. In order to clarify this, we rephrased the manuscript to emphasize these points. Equations 9 and 10 describe the soil redistribution flux by water in on hillslopes and in floodplains, respectively. The variable 'f' describes the fraction of the eroded sediment that is transported directly to the floodplains in the respective grid cell, while '1-f' describes the fraction of eroded sediment that remains on the hillslopes in the respective grid cell. The change of sediment storage in the floodplains and hillslopes of a grid cell as a result of erosion, deposition and transport are then together a representation for the net soil redistribution in that grid cell.

Changes in the manuscript: Line 140: "...sediment loss. Equation 1 can also be seen as a representation for the net soil redistribution flux, and can be approximated by the following as function of time:"

Line 231: "The modelling approach as presented by the equations above focuses on the net soil redistribution by separately modelling the main processes of soil redistribution, which are erosion, deposition and transport. In the following paragraphs we will show how this modelling approach performs for the Rhine catchment."

Answer part 2: We fully agree that a thorough validation of our approach in terms of net soil redistribution could provide valuable insights. However, reliable soil redistribution maps for large regions are very scarce. As suggested by the reviewer, we could use 137Cs-derived maps for net soil redistribution for Australia during the last $\sim$50 years, to validate our results. However, we feel that this is outside the scope of our study where

the focus is on millennial timescales. This could be done in a following-up study where we expand our modelling approach to other global catchments. It should be noted that 137Cs-derived maps represent soil redistribution for recent times, while in this study we focus on the soil redistribution for the last millennium. We have addressed this suggestion by adding the potential use of 137Cs-derived (and other approaches) on short-time scales as an outlook in the conclusions in the revised manuscript.

Changes in the manuscript: line 641: "...and future. The next steps in quantifying soil redistribution on the global scale will be to apply the sediment budget model on other large catchments or regions, particularly where data on net soil redistribution, sediment storage or yields exist."

Comment 2: "I think I understand why you needed to choose a grid resolution for your study. However, if the modelling is to be used in global ESM then it will need to be independent of grid resolution. I suspect that its application to other larger and flatter catchments would be represented by much of your model but perhaps not in the assumption that 5 arcmin is optimal for representing floodplain and hillslope. For example, Australia has some very large catchments which could mean that your chosen grid resolution may have no floodplain. I agree that a finer resolution would also not work in many regions but would work in other very steep, rugged terrain regions."

Answer: We agree that the model is currently limited to the resolution of 5 arcmin as it has been tested and calibrated for this specific resolution. We chose this particular resolution assuming that it includes both a hillslope and floodplain part for most of the Rhine catchment (and other catchments of similar size worldwide). We agree that if we want to make our model fully compatible with ESMs it would be good to make the model independent of grid resolution. However, we see our study as a first step into this direction. An issue that has to be addresses is the delineation of floodplain areas at global scales. Derivation of floodplains from soil properties and types is found to be insufficient. We are currently experimenting with deriving floodplains from Digital Elevation Models (DEMs). However, this is still work in progress. We will address this
issue in the conclusion part of the revised manuscript.

Changes in the manuscript: line 641: "...and future. The next steps in quantifying soil redistribution on the global scale are applying the sediment budget model on other large catchments, and validate the model with existing data on net soil redistribution, sediment storage or yields. Furthermore, in order to make the soil redistribution model better applicable on a global scale and to prevent conflict with the underlying assumption of the simultaneous presence of floodplains and hillslopes in each grid box, the model needs to be made independent of grid resolution."

Comment 3: "There is very little justification for the model parameter values in section 2.2 (page 8). You may be interested in considering for this manuscript and future work whether your approximation for f is consistent with the Multi-resolution Valley Bottom Flatness (MrVBF) data for Australia can be found on the link below. https://data.csiro.au/dap/landingpage?pid=csiro%3A5681 The land cover type and metrics are readily available Australia and presumably for precent slope so that might provide additional calibration / validation for your model. I'm not convinced about the use of average slope (as opposed to median) and which will be influenced by the resolution of the grid."

Answer: Thank you for this suggestion, we will consider this alternative approach in our further development of the model (see comment above). We can use this index to help identifying floodplains in a future study when applying the sediment budget model on a global scale (related to the previous comment). In addition to this we can then also use the data on land cover for Australia to validate our model. We will need to consider that due to the rapid changes of land use, recent maps are of limited use in our study that focuses on time-scales in the order of 1000 years.

Comment 4:"I think if scaling is to be an important part of the paper, as demonstrated by section 3.1 and abstract, then some background on Hoffman et al. (2013) needs to be included so that this topic in the manuscript is easier to comprehend."

**ESurfD**
Answer: We will add in the revised manuscript more background on the scaling relationships found by Hoffmann et al. (2013) and why this scaling is an important feature that needs to be reproduced with our model.

Changes in the manuscript: Line 285: "...catchment area. With these scaling relationships, for the first time, a direct comparison is made between the behavior of soil redistribution on hillslopes and in floodplains at large spatial scales. This is an essential difference between hillslopes and floodplains that sediment budget models like ours needs to capture in order to reliably simulate the spatial distribution of sediment.

Line 291:"...soil erosion. Hoffmann et al. (2013) indicated that with the estimated scaling coefficients (Eq.12 and Eq.13) even for large catchments (in the order of 105 km2) hillslopes store an equal amount of sediment as floodplains. They pointed out that this is a substantial sink that needs to be considered in sediment budgets of large catchments."

Comment 5:" I think it reasonable to qualify the extent to which this modelling can be used generally within ESMs to model soil redistribution by making explicit the contribution of soil redistribution by wind. In many global regions and drier more gusty phases in the past (and potentially the future), wind erosion may be considerably more important than water erosion. In other semi-arid regions the interplay between downslope erosion by water and removal by wind may cause a net soil redistribution to tend towards little change for some long period. In any case, I think it may be worthwhile for modellers interested in implementing your model that the wind erosion and dust emission component is important for soil redistribution. I accept that these processes may not be particularly relevant in the Rhine but note that they are in the sandy previous outwash plains of other parts of Germany and NW Europe."

Answer: We agree that soil redistribution by wind may play an important role for soil redistribution in many arid regions. Although we focus on soil redistribution by water in this study, generally considered to be the dominant erosion process at the global scale

(e.g. Quinton et al., 2010) we will add some sentences in the revised manuscript in the conclusions and outlook section on the importance of including soil redistribution by wind and other types of soil erosion in ESMs.

Changes in the manuscript: After line 641: "Finally to have a complete picture of the net soil redistribution and the feedbacks on the carbon and nutrient cycles, it is essential to model also other types of soil erosion, such as wind erosion (Chappell et al., 2015), tillage erosion (Van Oost et al., 2009) and gully erosion (Poesen et al., 2003)."

Comment 6: "I think the grammar and syntax of the manuscript need to be improved e.g., focus on consistent tenses. This might also be a good time to reduce the duplication of background material in the Introduction cf. start of Methods section. I think much of the technical justification might be moved from the Introduction to the start of the Methods section."

Answer: We will work on improving the syntax and grammar in the revised manuscript and try to avoid the duplication of material.

Comment 7: "References need some attention e.g., Oost or Van Oost."

Answer: We will correct for this mistake in the revised manuscript.

---

## Author Comment (AC2) · 21 Mar 2016

We would first like to thank Bertrand Guenet for his constructive comments. In this response we provide an answer to all the comments and the indicated changes will be applied in the revised manuscript.

Comment 1:"You used a modified version of the RUSLE equations without the support practice factor and then conclude that land use change is a driving factor of erosion. I think that you should discuss carefully how their conclusions would be impacted by the use of the support practice factor. In particular, how it could change the trends after the 1950's and during the middle age when animal traction was more and more used to plough."

Answer: The exclusion of the support practice factor, which represents the effect of

contouring, terracing, and subsurface drainage areas on erosion (Renard et al., 1997), may indeed impact the effect of land use change on erosion and the resulting sediment fluxes in regions with a long agricultural history, such as the Rhine catchment. However, our assumption that the factor equals unity in our study is consistent with a detailed assessment at the European scale where the average P factor for 2012 was estimated at 0.97 (Panagos et al., 2015). Furthermore, the study of Doetterl et al. (2012) showed that the S, R, C and K factors explain approximately 78% of the total erosion rates on cropland in the USA. This indicates that on cropland the L and P factors, which are related to agriculture and land management, contribute only for 22 % to the observed variability in erosion rates. Thus, although we neglect these factors in agricultural regions where they may play an important role, we expect that this does not affect the overall results of our study, such as that land use change is the driving factor of erosion. We will comment on the exclusion of L and P in the discussion.

Changes in the manuscript: Line 557: "... in cropland. Neglecting these factors in agricultural regions, where they may play an important role, results in an overestimation of the increases of soil erosion esp. during the 1950's, but we expect that this does not affect the overall trends. This assumption is supported by Doetterl et al. (2012), who shows that the L and P factors explain only up to 22% of the variability in water erosion rates on cropland in the USA."

Comment 2:"You used a modified version of RUSLE on non-croplands areas whereas this equation has been developed on croplands areas. I missed few sentences to justify that the use of RUSLE makes sense also for forest and grassland."

Answer: Indeed the original USLE model, the predecessor of RUSLE, was originally developed for cropland. However, as the model name already indicates, it is universal and can also be applied to forested and grassland areas. Model parameters for these land uses have been estimated using observational data and the model has been applied on a regular basis for the estimation of erosion in nature conservation areas, mine sites, forested areas and range- and grasslands (Dissmeyer, 1981; Millward and

Mersey, 1999; Lu et al., 2004). We will include this short explanation in the methods section of the revised manuscript.

Changes in the manuscript: Line 174: "... global scale. Although, RUSLE was originally developed for cropland, model parameters for other land cover types such as forest and grassland have also been estimated using observational data (Dissmeyer, 1981; Millward and Mersey, 1999; Lu et al., 2004).

Comment 3:" It would have been quite interesting to know the sensitivity of your approach to the inputs coming from the MPI-ESM using simulations coming from other ESMs. I am aware that this is asking a lot of additional work to redo everything using other ESMs outputs therefore adding just few elements in the discussion will be enough but at least it is important to mention it and to discuss how the uncertainties from the ESMs results might impact your conclusions."

Answer: Indeed, the sensitivity of our model to input data can be tested using data from other ESMs. We expect that the input data from other ESMs may significantly alter the trends in erosion rates and sediment fluxes for the last millennium. This is due to the fact that ESMs simulate climate and land cover in different ways.

Changes in the manuscript: Line 539: "... R factor remain. It is therefore also important to test the sensitivity of the sediment budget model with input data on precipitation and land cover from other ESMs."

Comment 4:"The discussion refers several times to the land use history but without enough references. Please document better this part."

Answer: We will add in the revised manuscript references to the studies of Hoffmann et al. (2007), Dix et al. (2005) and Kalis et al. (2003), who describe the long land use history of the Rhine catchment in more detail.

Minor comments: "L 133: You implicitly assume that movement of water during the flooding events do not induce erosion? If I understood well it should be clearly stated."

Answer: We only focus on rill and interrill erosion (which is indicated in the paper line 165), and not gully erosion or stream bank erosion that are the more extreme forms of erosion and related to flooding.

Changes in the manuscript: None

"L 151: I am not sure to fully understand what at and bt mean physically. Please clarify."

Answer: $a_\tau$ and $b_\tau$ in equation 3 are the adjustment parameters of our model relating the residence time to the flow-accumulation or catchment area. We will make this more clear in the revised manuscript.

Changes in the manuscript: Line 151: "...constants relating the residence time to the flow-accumulation or catchment area. Flowacc is the flow-accumulation and is defined as the number of grid cells...."

"L 355: Does it means that you use the same climate each year without inter-annual variability or do you repeat the sequence between 850 and 950 AD?"

Answer: We use the same yearly mean precipitation, temperature and the R factor, averaged over the 100-year period of 850-950AD (no inter-annual variability). In our study we calculated the R factor as an average over 100 year timeperiods starting from 850 AD and over a 50 year time period between 1950 and 2000.

"Fig. 4: Since it is scatter plot, you should fix the intercept to zero to have a better idea on how close to the 1:1 line the model is."

Answer: We will include the 1:1 line in the figure in the revised manuscript.

"Supplementary material I 44: If I understood well these parameters are fixed during the simulations? Why not use the stock of organic C predicted by the MPI ESM?"

Answer: Yes, the parameters to calculate the K factor, and the K factor itself are fixed during the simulations. We didn't use the C predicted by MPI-ESM because it is a very uncertain parameter of the model and it is therefore better to use the data from

GSCE. Also we assume that in the timeperiod of the last millennium the K factor will not change drastically in a way that it can change the erosion rates significantly.

---

## Editor Comment (EC1) · A.J.A.M. Temme (Editor) · 22 Mar 2016

I support resubmission of a revised paper, seeing that the main points from both reviews appear to be satisfactorily dealt with.